# Development of a Physicochemical Test Kit for On-Farm Measurement of Nutrients in Liquid Organic Manures

**Max-Frederik Piepel [1,2,*]** and **Hans-Werner Olfs [2]**

1 Division Plant Nutrition and Crop Physiology, Department of Crop Sciences, Georg-August-Universität Göttingen, Carl-Sprengel-Weg 1, 37075 Göttingen, Germany

2 Plant Nutrition and Crop Production, Faculty of Agricultural Sciences and Landscape Architecture, University of Applied Sciences Osnabrück, Am Krümpel 31, 49090 Osnabrück, Germany

* Correspondence: m.piepel@hs-osnabrueck.de; Tel.: +49-541-969-5135

**Abstract:** Optimised use of liquid organic manures (LOM) can reduce the consumption of mineral fertilisers and help reduce the emission of nutrients into nonagricultural ecosystems. To achieve this, farmers need to be able to measure the greatly variable nutrient composition of LOMs as accurately as possible on-farm. Since existing on-farm test methods either need to be precisely adapted to each LOM type or take a long time to perform, a test kit was developed to measure the nutrients of different LOM types within a short time. For the study, 619 LOMs (391 pig slurries, 139 cattle slurries, and 89 digestates) were collected from farms in northwest Germany and analysed in the laboratory for total N, ammonium, phosphorus, and potassium. The samples were analysed in parallel using the on-farm test kit consisting of ion-selective ammonium and potassium electrodes and an automatic moisture analyser to evaluate the comparability of the data. Each measurement could be performed in less than 15 min. Regardless of LOM type, regressions with an $R^2 > 0.9$ could be generated for total nitrogen, ammonium, and potassium, while the models for phosphorus were not as reliable.

**Keywords:** ammonium; biogas digestate; cattle slurry; pig slurry; potassium; phosphorus; total nitrogen





## 1. Introduction

In Germany, more than 187 million m³ of liquid organic manure (LOM) were applied to arable land in 2019. The three most important LOMs were cattle slurry with 94.7 million m³, biogas digestate with 62.8 million m³, and pig slurry with 27.6 million m³ [1]. Optimal use of these farm-based fertilisers can substitute costly mineral fertilisers, while plants can still be supplied with all relevant nutrients, and the environment can be protected [2]. The majority of mineral nitrogen fertilisers are produced via the Haber–Bosch process, which is extremely energy-intensive and releases large quantities of greenhouse gases [3,4]. Mineral phosphorus and potassium fertilisers are based on finite resources that must be transported over long distances [5], while liquid organic fertilisers are produced on the farm and mostly used locally. However, to substitute mineral fertiliser with organic manure, its nutrient content must be known as precisely as possible.

The nutrient compositions of LOMs vary widely. Therefore, farmers must have as much information as possible about the LOM nutrient content. Otherwise, there is a risk that too little or too much of a nutrient may be applied to a crop, which can result in either an undersupply of the plants or loss of nutrients in nonagricultural ecosystems. Nitrogen and phosphorus pose the greatest risks. Nitrogen can be emitted in gaseous form as $NH_3$ or $NO_2$ or leached into the groundwater as $NO_3^-$ [6,7]. Deposition of phosphorus via erosion or surface runoff into water bodies can lead to eutrophication [8,9]. For these reasons, it is imperative to determine the nutrient content of LOM before field application. Laboratory measurements are the most accurate method to evaluate nutrient concentrations in LOMs. However, they require representative sampling, which is associated with high effort for homogenisation of the storage containers. Several days usually pass before the results of

the measurements are available to the farmer. In addition, the costs for laboratory analyses are comparatively high, so often only one measurement per slurry tank is carried out per year. Therefore, there is a great need for methods with which nutrient concentrations can be determined directly by the farmer on site. One important methodological approach is to perform physicochemical quick tests.

Physicochemical quick tests can be used to measure various parameters of an organic fertiliser sample in a short processing time. Nutrient contents are then calculated based on the measured values. Since the 1970s, various regression models for cattle and pig slurries have been published, and their accuracy for calculating the concentrations has varied considerably depending on the LOM type, method, and nutrient in focus [10–12]. This is due to the different compositions of the LOMs. In general, the more similar the management systems of a group of LOM samples, the better the models fit [13]. However, the models can then only be applied to slurries from similar management systems; otherwise, errors occur in the calculation of nutrient concentrations. It is therefore desirable to have methods available that can be applied to a wide range of samples without restrictions.

Electrical conductivity is the most commonly used method to determine ammonium and potassium contents in LOMs [14–17]. However, since all ions present in the slurry influence the measurement [18], the interpretation of the measured data is not straight-forward. Usually, the more diverse the ion compositions of a sample set, the lower the fit is of the models, because all ions that are not in focus during the measurement are to be classified as interfering ions. One solution could be the selective measurement of ions in LOMs. For example, attempts have been made to selectively measure ammonium concentration with electrodes, but this has not been pursued in the last 20 years due to technical difficulties [19,20]. Since then, no further regression models for ion-selective electrodes have been published, although technology has improved, and ion-selective potassium electrodes are now commercially available.

Since large proportions of nitrogen and phosphorus are organically bound in LOMs, dry matter (DM) has been frequently used to derive these nutrient concentrations [21–23]. However, because the DM measurement can take up to 48 h [22], specific gravity (SG) has been used as a kind of "auxiliary parameter", because this parameter can be measured in a few seconds [24,25]. When SG is used to derive nitrogen and phosphorus concentrations in LOMs, it can be classified as a "double indirect" calculation (i.e., the concentrations are indirectly determined on DM, which is indirectly determined via an SG measurement), which may result in overall low fits of the regression models for different LOM types and management systems. A rapid direct determination of the LOM dry matter could avoid indirect derivation based on SG. However, no regressions using rapid DM determination to derive organic nutrients have been published to date.

This work aims to investigate whether it is possible to determine the ammonium, total nitrogen, total phosphorus, and total potassium concentrations of various LOMs using modern physicochemical measurement methods in as little as 15 min. We tested to what extent it is possible to create well-fitted regression models for very different LOM types and animal husbandry systems using ion-selective ammonium and potassium electrodes and an automatic moisture analyser.

## 2. Materials and Methods

### 2.1. Sample Collection and Sample Preparation

For the study, a total of 619 LOM samples (i.e., 391 pig slurries [232 fattening pig slurries, 110 sow slurries, and 49 piglet slurries], 139 cattle slurries [64 bull-fattening slurries and 75 dairy cattle slurries], and 89 digestates [based on various input materials]) were collected in Northwest Germany. Samples were taken by farmers following the standard procedures used to collect samples for laboratory testing. Ten litres of each LOM were homogenised for 3 min at 10,000 rpm at the experimental farm of Osnabrück University of Applied Sciences using a high-performance blender (Blender CB15VXE,

Waring Commercial, Torrington, CT, USA), filled into 500 mL containers and subsequently frozen at −18 °C.

### 2.2. Laboratory Measurements and Quick Test Methods

In an accredited laboratory (LUFA Nord-West; Hameln, Germany), nutrient concentrations were determined using the standard methods in Germany. Total nitrogen (TN) was analysed via a modified Kjeldahl method [26]. Ammoniacal nitrogen (AN) was determined by making up 5 g of the sample to 100 mL with 0.0125 molar calcium chloride solution. After filtration, the solution was measured based on a procedure using a continuous flow analysis method with photometric detection [27]. For total phosphorous (TP) and total potassium (TK), slurry samples were digested, and nutrient concentrations were determined by inductively coupled plasma optical emission spectrometry [28].

For the electrode measurements of ammonium and potassium, a measuring system from Mettler Toledo GmbH (Gießen, Germany) consisting of the ion meter "Seven2Go pH/Ion meter S8", the reference electrode "InLab Reference", and the temperature sensor "ATC NTC 30k Ohm" was used. For the ammonium measurement, the "DX218-NH$_4$ ISE" electrode was used, and for potassium measurement, the "DX239-K ISE" electrode was used. Sample preparation was identical for both ion measurements, i.e., 10 mL of MgSO$_4$ (250 mmol L$^{-1}$) was added to 10 g of LOM to increase the ionic strength of the solution and thus ensure a continuous ion flow to the ion-selective surface of the electrodes [20]. The solution was then filled up to 100 mL with distilled water to ensure that the ammonium and potassium concentrations in the sample solution were within the measuring range of the electrodes (maximum 1 mol L$^{-1}$). The samples were measured at 25 °C while stirring with a magnetic stir bar.

The MA35 infrared heated automatic moisture analyser from Sartorius AG (Göttingen, Germany) was used for the rapid determination of the dry matter in the LOM samples. For this measurement, 3 g of sample material was weighed using the scale integrated with the MA35. The sample was heated to 105 °C, and the weight loss of the sample due to evaporation was measured. As soon as no further weight loss was detected, the instrument automatically terminated the measurement and displayed the dry matter content in per cent. None of the measurements required more than 15 min.

### 2.3. Statistical Analyses

Simple descriptive statistical indicators (mean, minimum, and maximum) for the ammonium, total nitrogen, phosphorus, and potassium concentrations determined in the laboratory and by the quick tests, respectively, were calculated for the entire sample set and separately for the three LOM types. Simple linear regressions were created to estimate ammonium and potassium concentrations based on the electrode measurements as well as phosphorus concentrations based on the dry matter values of the moisture analyser. Multiple linear regression models were calculated to determine total nitrogen concentrations based on the ammonium electrode measurements and dry matter values determined by the moisture analyser.

To evaluate the quality of the models created, the coefficient of determination ($R^2$) and root mean square error (RMSE) were used:

$$R^2 = \frac{\sum_{i=1}^{n}(O_i - \overline{O}) \times \left(P_i - \overline{P}\right)}{\sqrt{\sum_{i-1}^{n}\left(P_i - \overline{P}\right)^2} \times \sqrt{\sum_{i-1}^{n}(O_i - \overline{O})^2}}$$

$$\text{RMSE} = \sqrt{\frac{\sum_{i=1}^{n}(Pi - Oi)^2}{n}}$$

where $P_i$ is the predicted value, $O_i$ is the observed value, $n$ is the number of observations, $\overline{O}$ is the average of the observed values, and $\overline{P}$ is the average of the predicted values. The model fit improves as $R^2$ approaches 1. The magnitude of the RMSE must be considered separately for each model because it depends on the magnitude of the measured value. When the RMSE is 0, there are no discrepancies between laboratory measurements and

quick tests. The greater the RMSE, the greater the deviations between the laboratory measurement and the quick test. All statistical calculations were performed with RStudio v1.4.1106 [29].

## 3. Results

The results of the laboratory measurements and the three quick tests are summarised in Table 1. The nutrient concentrations of the different LOM types (pig, cattle, digestate) differed strongly from each other, but there were also clear differences within one type of LOM. For total nitrogen as well as for ammonium, digestates show the highest average concentrations (5.43 and 2.83 kg m$^{-3}$). This is followed by pig slurry, with average concentrations of 3.66 and 2.33 kg m$^{-3}$ and cattle slurry with the lowest concentrations of 3.78 and 1.73 kg m$^{-3}$. In pig slurry, the ammonium concentration accounts for an average of 64% of the total nitrogen, in digestate for 52%, and in cattle manure for 46%. As for the phosphorus concentration, the digestates also have the highest average concentration with 0.87 kg m$^{-3}$, followed by the pig slurries with 0.71 kg m$^{-3}$, and the cattle slurries with 0.66 kg m$^{-3}$. The range of the pig slurries was wider than that of the digestates. While the pig slurry with the lowest concentration contained only 0.02 kg m$^{-3}$ phosphorus, the sample with the highest concentration contained 2.88 kg m$^{-3}$. This corresponds to a difference of 2.86 kg m$^{-3}$, while the differences were much smaller for the digestates with 1.69 kg m$^{-3}$ and the cattle slurries with 1.13 kg m$^{-3}$ (Table 1). The highest average potassium concentration was found again in the digestates with 4.64 kg m$^{-3}$. The second highest concentration of this nutrient was measured in the cattle slurries with 3.55 kg m$^{-3}$, and the lowest was in the pig slurries, with 2.37 kg m$^{-3}$. The digestates also had the largest range in potassium concentration, followed by the pig slurries and the cattle slurries.

**Table 1.** Total nitrogen, ammonium nitrogen, total phosphorus, and total potassium concentrations in liquid organic manures based on laboratory measurements (Lab) and quick tests.

| | Pig Slurries (n = 391) | | | | Cattle Slurries (n = 139) | | | | Digestates (n = 89) | | | |
|---|---|---|---|---|---|---|---|---|---|---|---|---|
| | **Min** | **Mean** | **Max** | **SD** | **Min** | **Mean** | **Max** | **SD** | **Min** | **Mean** | **Max** | **SD** |
| TN Lab (kg m$^{-3}$) | 0.50 | 3.66 | 9.08 | 1.59 | 0.90 | 3.78 | 5.69 | 0.86 | 2.60 | 5.43 | 11.10 | 1.35 |
| AN Lab (kg m$^{-3}$) | 0.20 | 2.33 | 5.06 | 0.91 | 0.31 | 1.73 | 2.88 | 0.46 | 1.10 | 2.83 | 7.90 | 1.05 |
| TP Lab (kg m$^{-3}$) | 0.02 | 0.71 | 2.88 | 0.62 | 0.11 | 0.66 | 1.24 | 0.24 | 0.13 | 0.87 | 1.82 | 0.25 |
| TK Lab (kg m$^{-3}$) | 0.33 | 2.37 | 5.83 | 0.98 | 0.75 | 3.55 | 5.97 | 0.92 | 1.25 | 4.64 | 8.22 | 1.05 |
| AN ISE (kg m$^{-3}$) | 0.19 | 2.05 | 4.99 | 0.80 | 0.39 | 1.60 | 2.66 | 0.43 | 0.86 | 2.57 | 6.70 | 0.90 |
| TK ISE (kg m$^{-3}$) | 0.27 | 2.16 | 5.64 | 0.87 | 0.72 | 3.13 | 4.96 | 0.79 | 1.26 | 4.02 | 7.11 | 0.98 |
| DM MA (%) | 0.15 | 3.42 | 15.31 | 2.84 | 1.00 | 7.12 | 11.52 | 2.30 | 1.48 | 6.60 | 10.14 | 1.73 |

AN = Ammonium nitrogen, TN = Total nitrogen, TP = Total phosphorus, TK = Total potassium. DM = Dry matter, EC = Electrical conductivity, ISE = Ion-selective electrode, MA = Moisture analyser.

The ammonium and potassium concentrations of the LOMs were also measured with ion-selective electrodes under close-to-farm conditions on the experimental farm of Osnabrück University of Applied Sciences. All mean values of the electrode measurements for ammonium are below the mean values of the respective laboratory measurements. For the pig slurries, a mean value of 2.05 kg m$^{-3}$ was measured with the electrode. For the digestates, a mean value of 2.57 kg m$^{-3}$ was determined, and for the cattle slurries, the measurement resulted in a value of 1.60 kg m$^{-3}$.

Similar results were obtained with the potassium electrode. As shown for the laboratory measurement, the electrode measured the lowest average concentration in the pig slurries with 2.16 kg m$^{-3}$. The cattle slurries had the second highest average value, with 3.13 kg m$^{-3}$. For the digestates, both the highest potassium concentration of 4.02 kg m$^{-3}$ and the largest difference from the laboratory value of 0.62 kg m$^{-3}$ were found. The dry matter of the pig slurries measured with the moisture analyser showed an average value of 3.42%; the range is largest for the LOM type. The digestates have the second largest

mean value with 6.60% and the smallest range. For cattle slurries, the highest mean value of 7.12% was detected.

Regression models for the determination of the four nutrients—ammonium nitrogen, total nitrogen, total phosphorus, and total potassium—were constructed using the three quick tests. For ammonium and potassium, data from the respective ion-selective electrode were used. Phosphorus concentrations were derived from the dry matter determined with the moisture analyser. For the total nitrogen model, both the data from the ion-selective ammonium electrode and the dry matter values from the moisture analyser were used. To check whether the methods are applicable regardless of the LOM type, a model was created for each nutrient for the respective type as well as a joint model for all 619 LOM samples (Table 2).

**Table 2.** Regression models for the different sample sets: laboratory measurements versus quick tests with coefficients of determination ($R^2$) and root mean square errors (RMSE) (significance level for all regressions $p < 0.001$).

| Nutrient | Manure Type | Equation | RMSE (kg m$^{-3}$) | $R^2$ |
|---|---|---|---|---|
| AN | Pig | y = 0.096 + 1.088 AN ISE | 0.262 | 0.92 |
| | Cattle | y = 0.096 + 1.022 AN ISE | 0.149 | 0.90 |
| | Digestate | y = 0.078 + 1.132 AN ISE | 0.268 | 0.93 |
| | All | y = 0.039 + 1.100 AN ISE | 0.247 | 0.93 |
| TN | Pig | y = 0.205 + 1.231 AN ISE + 0.273 DM MA | 0.354 | 0.95 |
| | Cattle | y = 0.379 + 1.190 AN ISE + 0.211 DM MA | 0.285 | 0.89 |
| | Digestate | y = 0.055 + 1.424 AN ISE + 0.261 DM MA | 0.438 | 0.89 |
| | All | y = 0.030 + 1.371 AN ISE + 0.242 DM MA | 0.391 | 0.94 |
| TP | Pig | y = 0.020 + 0.202 DM MA | 0.232 | 0.86 |
| | Cattle | y = 0.096 + 0.079 DM MA | 0.148 | 0.60 |
| | Digestate | y = 0.343 + 0.080 DM MA | 0.204 | 0.31 |
| | All | y = 0.145 + 0.122 DM MA | 0.350 | 0.54 |
| TK | Pig | y = 0.043 + 1.077 TK ISE | 0.298 | 0.91 |
| | Cattle | y = 0.039 + 1.125 TK ISE | 0.248 | 0.93 |
| | Digestate | y = 0.534 + 1.021 TK ISE | 0.315 | 0.91 |
| | All | y = 0.032 + 1.132 TK ISE | 0.302 | 0.94 |

AN = Ammonium nitrogen, TN = Total nitrogen, TP = Total phosphorus, TK = Total potassium. DM = Dry matter, EC = Electrical conductivity, ISE = Ion-selective electrode, MA = Moisture analyser.

The four models calculated with the measured values of the ammonium electrode have an $R^2 \geq 0.90$ for all types of manure (Table 2, Figure 1). The highest $R^2$ was found for the digestates with 0.93 (Figure 1D), followed by the models for the pig slurries with 0.92 (Figure 1C) and the cattle slurries with 0.90 (Figure 1B). The model based on all farm manures had an $R^2$ of 0.93 (Figure 1A). The RMSE for the model for the entire sample set with 0.247 was lower than the value for the model for the pig slurries, with 0.262.

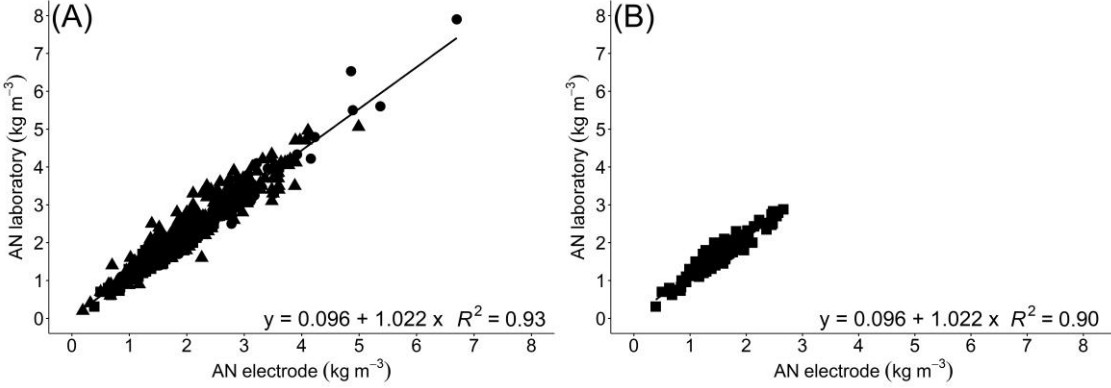

**Figure 1.** *Cont.*

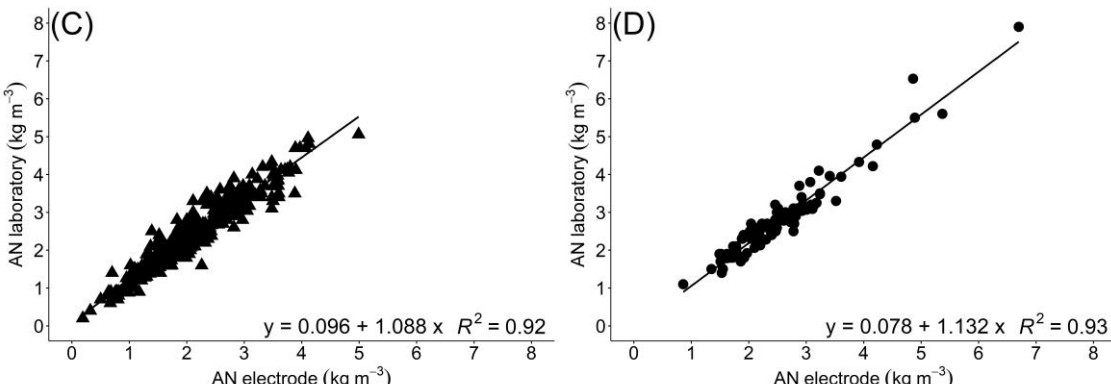

**Figure 1.** Relationships for the ammonium concentrations measured in the laboratory versus the ammonium concentrations measured with ion-selective electrode of the total sample set (n = 619): (**A**), cattle slurries (■) (n = 139); (**B**), pig slurries (▲) (n = 391); (**C**) and digestates (●) (n = 89); (**D**).

The total nitrogen model for all LOM samples had an $R^2$ of 0.94 and an RMSE of 0.391. The model for the pig slurries had a higher $R^2$ of 0.95 and a lower RMSE of 0.354. As mentioned earlier, the percentage of ammonium to total nitrogen was also highest for pig slurries. The model for the cattle slurries had a smaller $R^2$ of 0.89 than the overall model, but the RMSE for this model was also smaller. The model for the digestates had a smaller $R^2$ and a larger RMSE than the model for all farm manures (Table 2).

Figure 2 shows the regression models for the total phosphorus concentrations and the dry matter measured with the moisture analyser. It clearly shows that the $R^2$ of the models differed. The model of all farm manures had an $R^2$ of 0.54 and an RMSE of 0.350. As can be seen in Figure 2A, the deviations from the regression line became larger with increasing dry matter, and the total phosphorus concentration in pig slurries was overestimated based on the moisture analyser readings, while the values for cattle slurries and digestates were underestimated. The pig slurry model had a clearly higher $R^2$ of 0.86 and a lower RMSE of 0.232. However, Figure 2C also shows that the deviations from the trend line increased with an increasing dry matter for this data subset. The model for the cattle slurries had a higher $R^2$ of 0.60 and a lower RMSE of 0.148 than the overall model. However, higher deviations occurred with the increasing dry matter. The $R^2$ value for the digestate model was 0.31, the smallest value for all calculated models. The RMSE of 0.204 was lower than that of the overall model, but there were still large deviations for both low and high dry matter data (Figure 2D).

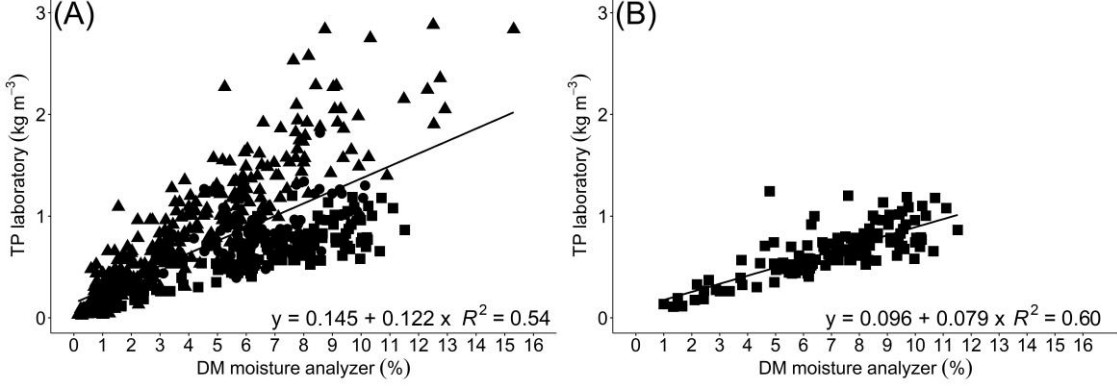

**Figure 2.** *Cont*.

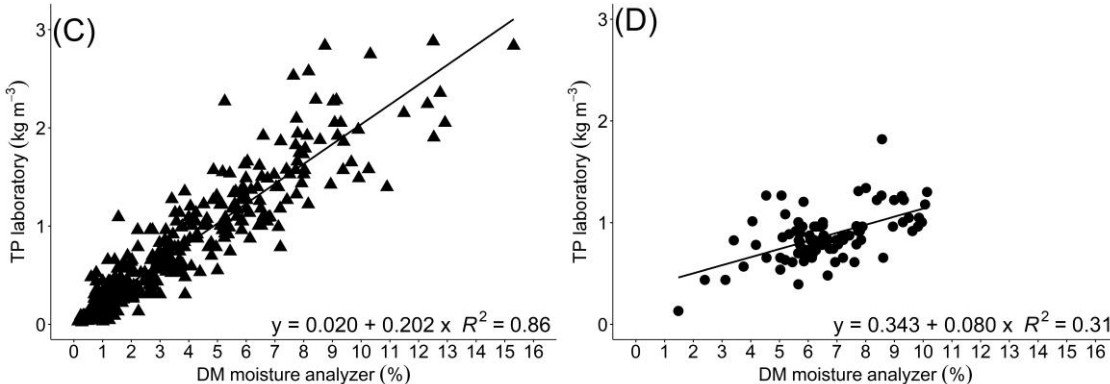

**Figure 2.** Relationships for the phosphorus concentrations measured in the laboratory versus dry matter measured with moisture analyser of the total sample set (n = 619): (**A**), cattle slurries (■) (n = 139); (**B**), pig slurries (▲) (n = 391); (**C**) and digestates (●) (n = 89); (**D**).

The four regression models based on the potassium electrode readings are shown in Figure 3. The model of all LOMs had an $R^2$ of 0.94 and an RMSE of 0.302. The model of the pig slurries (C) had a lower $R^2$ of 0.91 than the overall model and an identical RMSE of 0.302. The digestates model had a lower $R^2$ (0.91) than the model for all LOMs and a larger RMSE (0.315), while the cattle slurries model had a higher $R^2$ of 0.93 and a lower RMSE calculated with 0.248.

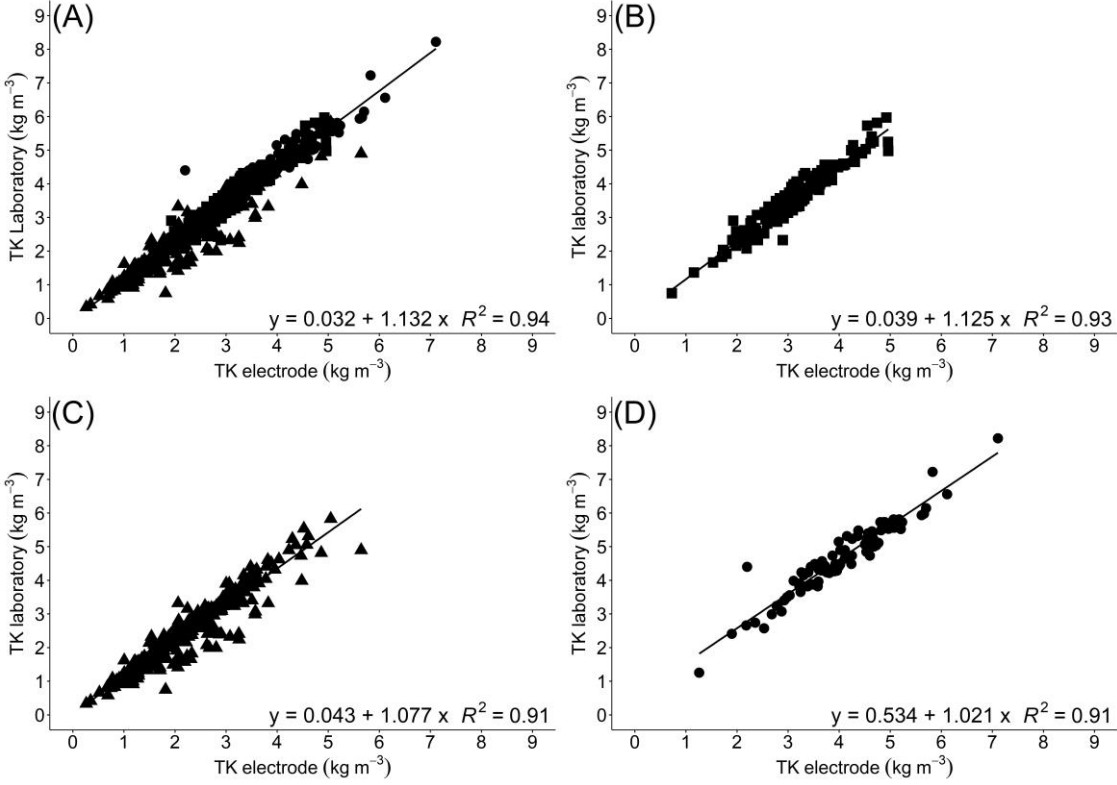

**Figure 3.** Relationships for the potassium concentrations measured in the laboratory versus the potassium concentrations measured with ion-selective electrode of the total sample set (n = 619): (**A**), cattle slurries (■) (n = 139); (**B**), pig slurries (▲) (n = 391); (**C**) and digestates (●) (n = 89); (**D**).

## 4. Discussion

The focus of this study was to evaluate a test kit that can be used to determine ammonium nitrogen, total nitrogen, total phosphorus, and total potassium on farms. To integrate the methods into the practical processes on a farm, a time window of 15 min was

targeted for the measurements and calculations of the nutrient concentrations based on the respective regression equations.

When using the standard method according to APHA [30], the DM measurement takes at least 1 h [13,24,25]. For larger sample volumes, the measurement may require up to 24 h [14] or even 48 h [22]. To reduce the time required for DM analysis, an electronic moisture analyser was used in this study. After weighing the sample, the weight loss in the drying process was automatically measured, and the process stopped when the weight remained constant. In our study, DM measurement did not take longer than 15 min for any sample. Since the electrode measurement can be carried out simultaneously during this period, it is possible to perform all measurements of the test set in the targeted timeframe.

### 4.1. Evaluation of the Ammonium Models

The values of the ion-selective ammonium electrode were used to create four models. Based on these models, it can be decided whether it makes sense to create a separate model for the individual LOM types or to calculate the ammonium concentration using the overall model.

Both the overall model and the models for the different LOM types have an $R^2 \geq 0.9$ and are thus at a similar level to the models for cattle and pig slurries published by Chescheir et al. [31] and Williams et al. [20]. In addition, it must be mentioned that the electrode used in our study did not drift during the measurement, and that linear rather than exponential models showed the best fits. Thus, large errors for the assessment of $NH_4-N$ (especially at higher concentrations in the LOMs), as mentioned by Williams et al. [20], do not occur (Figure 1).

The Chescheir et al. [31] model for cattle and pig slurries had a higher $R^2$ of 0.98 than the models in this study, but they used only 15 slurries in their model. Since they used an $NH_3$-sensitive electrode, they had to increase the pH of each sample to >12 with a strong alkaline leach. The sample preparation for the $NH_4+$ measurement used in this study does not include any substances potentially hazardous to farmers.

The durability of the membranes of the electrodes has also improved considerably in the meantime. While Byrne and Power [32] noted a deterioration in measurement quality after only 20 samples, all 619 samples in our sample set could be measured with the same membrane. However, when using modern ion-selective electrodes, the overall ionic strength must first be increased. In the second step, the LOM must be diluted before starting the measurement to ensure that the concentrations fit into the measuring range of the electrode. This is different for the determination of electrical conductivity. This measurement can be performed without any sample preparation. Due to this unproblematic measurement procedure, electrical conductivity is probably the physicochemical method with the most published regression models for ammonium determination in cattle slurries [15,33,34] as well as pig slurries [16,35,36]. However, since electrical conductivity correlates with the general ionic strength in LOMs [18,19] and ionic compositions differ between regions, manure type, and husbandry management systems [13], it is impossible to create well-fitted models for LOMs of different origins based on electrical conductivity data. This can be seen by looking at the models developed by Martínez-Suller et al. [14]. While the variability-limited model for the integrated farrow-to-finish slurries (n = 13) showed an $R^2$ of 0.95, the model of all slurries (n = 83) only had an $R^2$ of 0.82. For the calf slurries (n = 13), the $R^2$ for the total cattle manure (n = 49) was only 0.62. Variations in electrical conductivity were also found in the study by Singh and Bicudo [23]. While a model with an $R^2$ of 0.96 was calculated for the cattle slurries from the Monroe County region, the $R^2$ for the best model for the slurries from Barren County was only 0.61. Obviously, model fits differ regarding husbandry systems and regions.

Both electrical conductivity and ion-selective electrodes offer great potential for on-farm ammonium determination. When using data from electrical conductivity measurements, regression models adapted to the specific region and husbandry system must be calculated and updated when relevant changes (e.g., usage of other feeding ingredients)

occur. Measurements with ion-selective electrodes require careful sample preparation, which might be a bit time-consuming and challenging for farmers. However, with only one sample preparation procedure, ammonium and potassium concentrations can be measured, regardless of region, slurry type, and management system.

### 4.2. Evaluation of the Total Nitrogen Models

The total nitrogen in LOMs is composed of a mineral and an organic fraction. Since the mineral part consists essentially of ammonium, attempts have been made in the past to determine the total nitrogen concentration based on the electrical conductivity data [14,15,25]. On the other hand, most of the organic components are bound in the dry matter, so there are also regression models available to derive total nitrogen based on dry matter data [21,23,24]. To include both fractions in the calculation of the total N concentrations in LOMs, multiple regressions can be used, for example, with electrical conductivity and dry matter data [14,15,25]. The models developed in this study to derive the total nitrogen concentrations also include both fractions. Since the ion-selective ammonium electrode provided reliable data for the ammonium concentrations in LOMs independent of the slurry type and husbandry system, it was used to characterise the mineral fraction, while the dry matter measured with the moisture analyser was used to derive the organic fraction. The results show that the regression models fit as highly as models based on dry matter data determined in the laboratory. Marino et al. [15] calculated an $R^2$ of 0.91 based on electrical conductivity and conventionally determined dry matter for 93 dairy cow slurries. Martínez-Suller et al. [14] obtained an $R^2$ of 0.90 for the same parameters for 22 dairy cow slurries. This corresponds well to the fit of our model for the 139 cattle slurries. The best total N model for pig slurries was obtained by Martínez-Suller et al. [14] with a sample set of 40 farrowing sows ($R^2$ 0.89). This corresponds roughly to the model of Suresh and Choi [25] with 41 slurries and an $R^2$ of 0.88. The $R^2$ of the total N model for the pig slurries based on 391 samples in this study is even higher with 0.95 due to the use of the electronic moisture analyser to determine the dry matter within a maximum of 15 min. The overall model of all LOMs has an $R^2$ of 0.94. Obviously, the proposed test set using the ammonium electrode and the DM data from the automatic moisture analyser is thus promising for the determination of total nitrogen on-farm.

### 4.3. Evaluation of the Phosphorus Models

Up to now, there is no quick physicochemical test available that can measure phosphorus concentration in liquid organic manures directly on the farm. Therefore, phosphorus concentrations must be derived indirectly from other parameters. Since most of the phosphorus is organically bound, dry matter, which can be quickly determined with the moisture analyser, was used to build the models in this study. The P models for the different LOM types differ more clearly from each other than the models for the other nutrients. Although the pig slurry model has an $R^2$ of 0.86, the deviations from the trend line increase with increasing dry matter (Figure 2). The cattle slurry model has a slightly higher $R^2$ of 0.60 than the overall model, while the digestate model, with an $R^2$ of 0.31, is the worst-fitting model in this study.

The differences in the accuracy of the model fit are also found in the literature. Mostly oven drying according to APHA [30] is used, and this procedure can take up to 48 h [22]. Nevertheless, the cattle model in our study has an $R^2$ of 0.60, which is comparable to models from other studies with similar variable sample sets. Martínez-Suller et al. [14] obtained an $R^2$ of 0.62 for their combined cattle slurry data set (n = 49), and Marino et al. [15] an $R^2$ of 0.62 for their dairy cattle slurry data set (n = 93). Singh and Bicudo [23] showed that higher model fits are possible. They collected cattle slurry samples from different regions and determined the specific gravity of the slurry to derive dry matter data to calculate phosphorus concentrations. While their Hart County sample set only had an $R^2$ of 0.58, the Monroe County sample set had an $R^2$ of 0.99. This shows that regional differences in model fits can occur.

For pig slurries, limiting sample variability by selecting certain husbandry systems may result in higher model fits. While the model of the total sample set of pig slurry (n = 83) by Martínez-Suller et al. [14] had an $R^2$ of 0.37, $R^2$ increased to 0.83 by restricting variability using only finisher pig slurries (n = 30). Even better results were achieved by Zhu et al. [24]. They restricted their sample origin by taking samples from just nine farms, adjusted different dry matter values by adding water, and then were able to calculate an $R^2$ of 0.99.

The derivation of phosphorus concentration in digestates has not yet been published. Nevertheless, it can be assumed that it might also be possible to increase the $R^2$ of the models for digestates if the variability is restricted. This could be achieved, for example, by selecting specific digestate types (e.g., biogas plants just with maize as feeding substrate) or even by creating models for individual farms.

Overall, the model fits of the phosphorus models in this study are not satisfactory compared to the models for the other nutrients. To enable farmers to derive P concentrations in LOMs based on dry matter measurements, models need to be created for individual regions and husbandry systems. Piepel et al. [37] presented a smartphone app to make this possible in the future, i.e., farmers can create their own models for their region or even for their own farms.

### 4.4. Evaluation of the Potassium Models

As with ammonium nitrogen, most of the potassium in liquid manure is present in dissolved form. Therefore, electrical conductivity is the most commonly used parameter to derive this nutrient [14,15,38]. However, $K^+$ ions are only the second largest cation fraction after $NH_4^+$ ions in most organic manures [39]. Thus, $NH_4^+$ ions have a greater influence compared to $K^+$ on the electrical conductivity in most cases [18]. The potassium models are therefore usually worse fitted than the ammonium models. For example, the potassium model of Marino et al. [15] with 38 cattle slurries had an $R^2$ of only 0.30, whereas the ammonium model for the same sample set had an $R^2$ of 0.76. As with ammonium, attempts have been made in the past to increase the model fit by selecting certain husbandry systems. The combined cattle model of Martínez-Suller et al. [14] had an $R^2$ of only 0.27, and the model of the dairy cow slurries had an $R^2$ of 0.64. For the pig slurries, they obtained the highest $R^2$ (0.84) for their sample set by only focusing on the 13 integrated farrow-finish slurries. The $R^2$ for the total pig slurries set, however, was only 0.52. Moral et al. [40] calculated an $R^2$ of 0.82 based on 36 pig slurries. Up to now, no physicochemical models for the determination of the potassium concentration in digestates have yet been published. The ion-selective potassium electrode could probably provide the highest fitted models, as all models of this study have an $R^2 > 0.9$. The overall model of all LOMs shows that no restrictions on sample variability are necessary.

## 5. Conclusions

With the test kit consisting of ion-selective electrodes and an automatic moisture analyser, it is possible to perform all the necessary measurements to assess the nutrient concentrations in LOMs on-farm and in a fast and uncomplicated manner. The ion-selective electrodes provided very well-fitted models for ammonium and potassium determination independent of the region of origin, LOM type, or husbandry system and are therefore more universally applicable than the measurement of electrical conductivity. With the moisture analyser, the measuring time for determining dry matter can be reduced to less than 15 min. Together with the ammonium electrode, it was possible to calculate well-fitted models for the total nitrogen concentration. It should be emphasised that the sample type (i.e., pig, cattle, digestate) does not affect the prediction quality of the model. The goodness of fit of the phosphorus models based on the dry matter was significantly lower than those of the other nutrients. In order to determine phosphorus concentrations in LOMs directly on the farm in the future, better fitted dry matter models need to be created. This will require determining how to limit variability in the sample set and how this will improve

model fit. Alternatively, methods for direct measurement of phosphorus concentration could be developed.

**Author Contributions:** Conceptualization, M.-F.P. and H.-W.O.; methodology, M.-F.P.; software, M.-F.P.; validation, M.-F.P. and H.-W.O.; formal analysis, M.-F.P.; investigation, M.-F.P.; resources, M.-F.P. and H.-W.O.; data curation, M.-F.P.; writing—original draft preparation, M.-F.P.; writing—review and editing, M.-F.P. and H.-W.O.; visualization, M.-F.P.; supervision, H.-W.O.; project administration, H.-W.O.; funding acquisition, H.-W.O. All authors have read and agreed to the published version of the manuscript.

**Funding:** This work was supported by the Deutsche Bundesstiftung Umwelt (grant number 33702/01).

**Institutional Review Board Statement:** Not applicable.

**Data Availability Statement:** The data presented in this study are available on request from Hans-Werner Olfs.

**Acknowledgments:** We acknowledge support by the Open Access Publication Funds of Göttingen University.

**Conflicts of Interest:** The authors declare no conflict of interest.

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
