# Peer review of "Development of a Physicochemical Test Kit for On-Farm Measurement of Nutrients in Liquid Organic Manures"

_agriculture, doi:10.3390/agriculture13020477_

Round 1

Reviewer 1 Report

The presented results of the study are relevant and applicable to agriculture.

In your conclusions you have indicated that it is necessary to create better-fitted dry matter models for phosphorus determination. Do you plan to do such phosphorus modeles in your future research? It would be appropriate to clearly indicate what areas of future research you plan to do.

Author Response

Dear reviewer,

Thanks a lot for the constructive comment on our manuscript “Development of a physicochemical test kit for on-farm measurement of nutrients in liquid organic manures”. We have seriously considered the comment and have taken it into account in revising the manuscript. The authors' answer is written in red.

The presented results of the study are relevant and applicable to agriculture. In your conclusions you have indicated that it is necessary to create better-fitted dry matter models for phosphorus determination. Do you plan to do such phosphorus models in your future research? It would be appropriate to clearly indicate what areas of future research you plan to do.

Previous research has shown that better-fitted dry matter models for phosphorus determination than those we have presented are possible (lines 355-372). In one of our papers (Piepel et al. 2022), we have developed a mobile app that enable farmers to select (or even create) better fitted models. We have also shown that grouping pig slurries into production stages (e.g. sows, piglets, fattening pigs) is not sufficient to automatically increase model fit. In the future, we want to explore the effects of sample variability on the performance of regression models and how farmers can use these models as easily as possible.

We have adapted the conclusions accordingly.

Please do not hesitate to contact us if you have any further questions.

Reviewer 2 Report

A physical-chemical reagent kit for field determination of nutrients in liquid organic manures was proposed in this manuscript. The independent and general model for different reagent detection of multi-category organic manures were developed, and compared with the research results of peers. The experimental design is reasonable and the data analysis is sufficient, but the following minor revisions are required.

1. Please correct the following statement that cannot be automatically connected.

line 23-24: while the models for phosphorus were not as reliable. Reliable on-site methods for characterising LOMs are thus available to farmers."

2.Please clarify the following contradictory statements:
line43-44: the following formulas were used to determine the coefficient of determination (R²)
line 53-54: All statistical calculations were performed with R [29]

Author Response

Dear reviewer,

Thanks a lot for the constructive comments on our manuscript “Development of a physicochemical test kit for on-farm measurement of nutrients in liquid organic manures”. We have seriously considered the comments and have taken them into account in revising the manuscript. The authors' answers are written in red.

  1. Please correct the following statement that cannot be automatically connected.

line 23-24: while the models for phosphorus were not as reliable. Reliable on-site methods for characterising LOMs are thus available to farmers."

The text has been adapted.

2.Please clarify the following contradictory statements:
line43-44: the following formulas were used to determine the coefficient of determination (R²)
line 53-54: All statistical calculations were performed with R [29]

We have separated the formulas and the software used more clearly in the text.

Please do not hesitate to contact us if you have any further questions.

Max Piepel

Reviewer 3 Report

Dear authors,

I have some minor comments on the manuscript. I enjoyed reading it.

L110-111: what type of spectroscopy?

L154: add the version number of R.

There are no figures for AN?

L258: what do you mean by evaluation of the data?

Author Response

Dear reviewer,

Thanks a lot for the constructive comments on our manuscript “Development of a physicochemical test kit for on-farm measurement of nutrients in liquid organic manures”. We have seriously considered the comments and have taken them into account in revising the manuscript. The authors' answers are written in red.

L110-111: what type of spectroscopy?

The measurement was carried out with a photometer. The text was changed accordingly.

L154: add the version number of R.

The version has been added.

There are no figures for AN?

The models for AN are shown in Figure 1. However, there are no graphs for TN as the four models presented are multiple regression models (Table 2). We would not like to insert three-dimensional graphs, as the form of representation would not contribute to an easy understanding of the models.

L258: what do you mean by evaluation of the data?

Within the 15-minute period, the nutrient concentrations should be calculated with the regression equations. This is possible with a calculator or the mobile app we presented in Piepel et al. 2022.

Please do not hesitate to contact us if you have any further questions.

Max Piepel